# Spicy and Aromatic Plants for Meat and Meat Analogues Applications

**DOI:** 10.3390/plants11070960

**Published:** 2022-04-01

**Authors:** Romina Alina Marc (Vlaic), Vlad Mureșan, Andruţa E. Mureșan, Crina Carmen Mureșan, Anda E. Tanislav, Andreea Pușcaș, Georgiana Smaranda Marţiș (Petruţ), Rodica Ana Ungur

**Affiliations:** 1Food Engineering Department, Faculty of Food Science and Technology, University of Agricultural Science and Veterinary Medicine Cluj-Napoca, 3-5 Calea Mănăştur Street, 400372 Cluj-Napoca, Romania; romina.vlaic@usamvcluj.ro (R.A.M.); crina.muresan@usamvcluj.ro (C.C.M.); anda.tanislav@usamvcluj.ro (A.E.T.); andreea.puscas@usamvcluj.ro (A.P.); georgiana.petrut@usamvcluj.ro (G.S.M.); 2Department of Rehabilitation Iuliu-Haţieganu, Faculty of General Medicine, University of Medicine and Pharmacy, 8 Victor Babes Street, 400012 Cluj-Napoca, Romania; rodica.ana.ungur@gmail.com

**Keywords:** herbs, essential oils, aroma compounds, antioxidant activity, antibacterial activity, bioactive compounds

## Abstract

Aromatic and spicy plants are an important factor that contributes not only to improving the taste of meat, meat products, and meat analogues, but also to increasing the nutritional value of the products to which they are added. The aim of this paper is to present the latest information on the bioactive antioxidant and antimicrobial properties of the most commonly used herbs and spices (parsley, dill, basil, oregano, sage, coriander, rosemary, marjoram, tarragon, bay, thyme, and mint) used in the meat and meat analogues industry, or proposed to be used for meat analogues.

## 1. Introduction

Spicy and aromatic plants have been used in human consumption for thousands of years (since around 5000 BC). Initially, they played an important role in primary care, being used as therapeutic agents in the treatment of various diseases; however, wider applications are reported today [1]. Over time, spicy and aromatic plants began to be used around the world in various foods to flavor them, but also for preservative purposes. These plants are considered an untapped reservoir of valuable substances, also called phytochemicals, phytogenic, phytobiotics, botanicals or spices, although they are not established as essential ingredients [1,2,3,4].

Meat is known to be an important source of protein, essential amino acids, vitamins, and minerals. However, most of the meat worldwide is processed. After processing, the meat becomes more perishable and sensitive to oxidation. To improve these characteristics, as well as the aroma, aromatic and spicy plants with aromatizing roles and natural antioxidants are used [5,6]. Synthetic chemicals are also used, but consumers prefer natural antioxidants due to the possible long-term toxic effects of synthetic substances [7].

The growing population around the world has led to the need to increase the number of protein-containing products. Meat and meat products are the most common sources of high protein, but these sources are no longer able to meet all the needs of consumers: an increasing amount is needed, and for a part of the population, these products are not recommended for certain diseases. Along with this need, the interest in meat analogues has risen considerably [8]. The demand for these vegetable meat alternatives is growing, because they have benefits for consumers, but also for the planet, and are recognized as sustainable protein sources. A vegetable-based diet has been shown to reduce the risk of cardiovascular disease, diabetes, high blood pressure, and mortality [9].

To make these food products tasty, and to have a pleasant appearance, whether referring to meat products or meat analogues, we use herbs and spices. They are used not only for their flavor, but also for the benefits they bring to finished products and the benefits they bring to consumers. In the meat and meat analogues industries, the most commonly used aromatic and spicy plants are parsley, dill, basil, oregano, sage, coriander, rosemary, marjoram, tarragon, bay, thyme, and mint [2,10,11,12,13,14,15,16,17,18,19,20,21,22,23,24,25,26,27,28]. They contain chemicals such as polyphenols, flvonoids, quinics, polypeptides, and alkaloids, or their oxygen-substituted derivatives. Some of these substances can act synergistically and improve bioactivity. Additionally, these bioactive compounds have therapeutic value, such as antioxidant and antiseptic activity [1]. Thus, the active components of these plants might have the ability to reduce the risk of cancer, cardiovascular disease, respiratory disease, and stomach or inflammatory disorders, and reduce oxidative stress. They also contain antimicrobial compounds, which delay microbial growth in food [1,4]. This review aims to gather recent information on spicy and aromatic plants used to prepare meat and meat alternatives.

## 2. Spicy, Aromatic Plants and Their Applications in Meat and Meat Analogues

The aromatic and spicy plants regularly used in meat preparations and meat analogues are parsley, dill, basil, oregano, sage, coriander, rosemary, marjoram, tarragon, bay, thyme, and mint. Different parts of plants are used, such as stems, seeds, or leaves, in different forms, including extract powder, essential oils, ground leaves, herbal dust, water extract, or powder extract. They are used in different amounts depending on the type of product to which they are added, and have flavoring, coloring, antimicrobial, antioxidant effects, as shown in Table 1.

## 3. Bioactive Compounds and Antibacterial and Antioxidant Activity of Spicy and Aromatic Plants in Meat

The consumption of fresh and processed meat has an indisputable value for diet as a source of proteins and micronutrients, but in 2016, an International Agency for Research on Cancer (IARC) working group classified processed meat as “carcinogenic to humans” and red meat as “probably carcinogenic to humans for colorectal cancer”. On the other hand, fresh and processed meat is an key part of the Western diet, associated with chronic metabolic inflammation and an important group of chronic diseases: obesity, hyperlipidemia, diabetes, gout, high blood pressure and degenerative neurological diseases, including dementia. In this context, it is very important to associate meat and its derivatives with compounds that can counteract their possible negative effects. 

Spices can be important factors that contribute not only to improvements in meat and meat product savor, but also to increasing the nutritive value of meat and derivatives, and counteracting their metabolic disadvantages. It has been demonstrated that plants with green leaves utilized as spices for meat and derivatives have important antioxidant and anti-inflammatory effects, antidiabetic, antimicrobial and antimutagenic actions, associated with their chemical composition, i.e., rich in polyphenols, carotenoids, and terpenoids. The use of synthetic antioxidants is restricted due to their carcinogenicity; therefore, natural antioxidants derived from plants, including aromatic plants, are recommended for use in the food industry. The prevalence of digestive disorders in Western populations is currently increasing. Culinary spices used in meat preparations can stimulate digestive processes by increasing bile and digestive enzyme production, and modulating the structure and function of gut microbiota. On the other hand, the antioxidant, anti-inflammatory, antimicrobial, antifungal, and antimutagenic effects of spices protect the digestive system from cancer, gastritis and ulcers, periodontitis, and colitis. Recently, many spices used in meat processing have been proposed as alternative treatments for SARS-CoV-2-infected patients due to their anti-inflammatory properties that can be potentially efficient to combat a cytokine storm [29].

Due to their ability to prevent and slow down the rate of lipid oxidation in food systems, aromatic herbs and their derivatives are potential natural alternative sources for antioxidants and synthetic preservatives. They can be added directly to the product (in fresh or dried states), its constituents can be extracted and added in the form of essential oil (the addition is limited by the intense aroma) or extracts, and in the combination of different plant extracts. The last method can lead to superior antioxidant activity due to the synergistic action of the compounds. Natural antioxidants have the effect of reducing the formation of cytotoxic compounds during the thermal processing of food (for example, in meat) [30].

**Parsley** (*Petroselinum crispum* Hoffm) stems contain many bioactive compounds such as carotenoids, including *β*-carotene, neoxanthin, violaxanthin, lutein, and glycoside apiose. The leaves have a high content of vitamins (K, A, C), folate, niacin, choline, pantothenic acid and also *β*-carotene, lutein. Parsley is used to improve appetite and alleviate indigestion, flatulence and spasms, and may prevent stomach ulcers [31]. Stem and leaf extracts of *Petroselinum crispum* have been identified to exert antioxidant, anti-inflammatory, and antiplatelet activities, and have protective effects against hyperuricemia and hyperglycemia, brain, heart, and liver diseases [32].

It proved to inhibit the bacterial growth of *Bacillus subtilis*, involved in the pathogenesis of digestive anastomotic leak, and of the pro-inflammatory proteobacterium *Escherichia coli*, involved in Crohn’s disease, whose overgrowth is facilitated by Western diets [33,34,35].

According to Zhang et al. [36], parsley essential oil exhibits antioxidant activity due to the compounds myristicin (phenylpropene) and apiol (phenylpropanoid). The flavonoids isorhamnetin, apigenin, quercetin, luteolin, and chrysoeriol represent the predominant compounds in cell suspension cultures of parsley [32]. Parsley apigenin (yellow color) can be used as a pigment in human and animal nutrition [1].

Wong and Kitts [37] studied the in vitro antioxidant activity by DPPH inhibition of free radicals, ion-chelating, and hydroxyl radical assays from aqueous and methanolic extracts prepared from parsley leaves and stems. The methanolic extracts from the leaves showed a significant (*p* < 0.05) radical scavenging activity, attributed to the total phenolic content, whereas the chelating activity of the ferrous ions was significantly (*p* < 0.05) higher in methanol extracts from the stem.

Phenolic compounds are the main class of plant compounds that contribute to their antioxidant capacity [38].

Due to its high nitrate content, different concentrations of parsley extract powder have been used in the formulation of sausages (V3: 1.07 g, V4: 2.14 g and V5: 4.29 g parsley extract/kg meat), in comparison, from a sensory point of view, with traditional sausages with nitrites (V1) and without nitrites (V2). The evaluators did not observe any significant differences between samples V1, V3, V4, and V5 in terms of red color intensity, although sample V2 showed intense grey color (due to lack of nitrate). Parsley extract powder did not show an influence on taste and aroma and no significant differences were found between the sausage variants. Among sausages produced with parsley extract powder, the general acceptance for sample V3 was reduced, due to the lower concentrations of parsley extract, resulting in lower levels of nitrites and lower nitrosomyochromogen contents formed. Thus, depending on the sensory parameters evaluated, samples V4 and V5, produced with 2.14 g and 4.29 g of parsley extract/kg meat, respectively, had characteristics similar to those of the traditional counterpart treated with nitrate [10].

**Rosemary** (*Rosmarinus officinalis* L.) leaves are rich in compounds with health-promoting properties for a variety of diseases. The main bioactive compounds are phenolic diterpenes—carnosic acid and carnosol; triterpenes—oleanolic, betulinic, and ursolic acids; flavonoids—hesperidin, homoplantaginin, cirsimaritin, genkwanin, gallocatechin, nepetrin, 6-hydroxyluteolin-7-glucoside, luteolin-3′-glucuronide, and luteolin-3′-*O*-(*O*-acetyl)-*β*-d-glucuronide; and typical compounds—rosmarinic acid, rosmadial, and rosmaridiphenol [39]. They confer antioxidant, anti-inflammatory, antihyperlipidemic, hepatoprotective, renoprotective, antithrombotic, antinociceptive, antidepressant, antimicrobial, and anticancer properties. In traditional medicine, *Rosmarinus officinalis* L. has been used to treat gastrointestinal, hepatic, cardiovascular, nervous, respiratory, genitourinary, and skin disorders [40,41]. In experimental models, rosemary extracts improved body weight control, total cholesterol level, and atherogenic index, cardiac remodeling after myocardial infarction, brain tolerance to artificially induced ischemia, and protection against rupture of the blood–brain barrier [42]. Anticancer properties were demonstrated in several cell-line models, for esophageal squamous cell carcinoma (KYSE30), gastric adenocarcinoma (AGS), epithelial colorectal adenocarcinoma (CaCo-2), breast adenocarcinoma (MCF-7), cervical adenocarcinoma (HeLa), lung carcinoma (A549) histiocytic lymphoma (U-937), and human melanoma (A375) [43]. Rosemary extracts also exert antifungal, antiviral, and antibacterial activities [41].

The stability, antioxidant and antimicrobial activity of rosemary essential oil introduced into meat and meat products can be improved by encapsulating it in a nanogel of chitosan and benzoic acid [44]. Krkić et al. studied the incorporation of oregano essential oil into a chitosan coating, which reduced lipid oxidation, and contributed to the formation of a smaller amount of aldehydes and superior sensory properties in dry fermented sausages [45].

Rosemary extract (0, 250, 500, and 750 mg/kg) combined with sodium nitrite (40, 80, and 120 mg/kg) was used to obtain liver pâté and to study color stability, lipid oxidation, and concentrations of ascorbic acid, α-tocopherol, and carnosic acid. Regardless of the added dose of rosemary extract, it had a significant effect in reducing lipid oxidation and maintaining high levels of antioxidants, while having no effect on color stability. The concentration of carnosic acid increases with the dose of rosemary extract. Low doses of sodium nitrite (80 mg/kg) can be used without adversely affecting color stability, forming significantly lower nitrite concentrations and slightly reduced lipid oxidation values, while the use of rosemary extract helps maintain lipid stability [46].

Rosemary and marjoram essential oils have been added in different doses to pork sausages in a study conducted by [47]. The results demonstrated the protective effects of essential oils against oxidation. In addition, the increase in TBARS values and the loss of red color was prevented and compared with samples containing synthetic antioxidants; samples with essential oil obtained similar or better results. Mohamed and Mansour evaluated the antioxidation efficiency of rosemary and marjoram essential oils (200 mg/kg) added to frozen beef patties and stored for 3 months at −18 °C. Essential oils have been shown to be effective against oxidation. The results of a sensory analysis showed that the addition of essential oils had a positive effect on the samples, being highly appreciated by evaluators [48].

According to the Food Safety and Inspection Service (FSIS) Directive 7120.1: “Safe and suitable ingredients used in the production of meat, poultry, and egg products”; rosemary extract is the most commonly used natural antioxidant in the meat industry, with it explicitly being allowed for use as a component of an antioxidant mixture [49]. The antioxidant properties of rosemary are due to the presence of phenolic diterpenes, namely carnosic acid and carnosol, which act as hydrogen donors in the reaction with free radicals [46]. In the case of sage, the antioxidant capacity is due to the presence of phenolic diterpene (epirosmanol, carnosol, and carnosic acid) [50]. Rosemary and sage extracts can provide antioxidant species in both the polar and non-polar phases of a food product. For example, carnosic acid is on the lipophilic end of the scale, and rosmarinic acid is on the hydrophilic end. Carnosic acid is a superstoichiometric antioxidant because it can act repeatedly as a reducing agent by donating hydrogen atoms through phenolic compounds [51].

**Sage** (*Salvia officinalis* L.) leaves are rich in terpenes, anthraquinone, and flavonoids with antioxidant, anti-inflammatory, and antimicrobial effects. In sage essential oils, the main components are camphor, 1,8-cineole, *α*-thujone, *β*-thujone, borneol, and viridiflorol [52]. In traditional medicine, *Salvia officinalis* has been used to treat mild dyspepsia, ulcers, and gout. The German Commission E has also accepted the use of *Salvia officinalis* for dyspepsia [53].

It was reported that drinking sage tea prevented the initiation phases of colon carcinogenesis in an experimental rat model [54]. Clinical trials reported memory-enhancing and antidementia benefits in healthy adults or patients with Alzheimer’s disease [55], and hypoglycemic and hypolipemic effects of *Salvia officinalis* leaves in patients with diabetes and hyperlipidemia and in healthy volunteers [53].

Sage may produces several types of phenolic species as opposed to rosemary, especially in the production of flavonoids and other phenolic derivatives, although rosemary produces higher amounts of carnosic acid and other diterpenoids related to it [51]. In a study by Kontogianni et al., they showed that rosemary extract is twice as rich in diterpornoid and phenolic compounds and contains about 2.7 times more carnosic acid than sage extract [56].

**Dill** (*Anethum graveolens* L.) seeds used in meat products and derivatives are rich in essential oil, of which the major compounds are carvone, limonene, and camphor, characterized by important antioxidant activity [57]. Antioxidant activity also characterizes the flavonoids quercetin and isoharmentin isolated from *Anethum graveolens* L. seeds, which can help to prevent peptic ulcers. This effect has been verified in experimental models in which aqueous and ethanolic *Anethum graveolens* L. extracts had mucosal protective and antisecretory effects, similarly to high-dose sucralfate [58].

The essential oils found in seeds are carminative, improve appetite, aid digestion, and relieve intestinal spasms. d-limonene is a monoterpene that dissolves cholesterol-containing gallstones. It is chemopreventive and has chemotherapeutic activities [59]. In experimental models, dill seed extracts suppressed hyperlipidemia induced by a high-fat diet [60] and had inhibitory effects on hepatic carcinoma cells [61]. 

Extracts from dill (*Anethum graveolens*) obtained in organic and conventional agriculture were prepared in n-hexane, dichloromethane (CH_2_Cl_2_), ethyl acetate (EtOAc) and ethanol (EtOH), and the radical scavenging activity (RSA) at 2000 µg mL^−1^ has been studied by DPPH (2,2-Diphenyl-1-picrylhydrazyl), DMPD+ (N,Ndimethyl-p-phenylendiamine), and NO (nitric oxide) methods. Ethanol extracts (both conventional and organic agriculture) had better inhibitory effects, and the NO radical scavenging activity was particularly noted (78.49 ± 1.86% for conventional agriculture and 71.86 ± 5.41% for organic agriculture). No significant differences were observed between organic and conventional agriculture extracts in the RSA tests. Ferric-ion-chelating capacity and phosphomolybdenum-reducing antioxidant power (PRAP) assays were also studied; dichloromethane (CH_2_Cl_2_) extracts had the greatest ferric ion chelation effect (74.34 ± 1.40%), and PRAP values from both extracts had better values than the rest of the samples [62].

**Oregano** (*Origanum vulgare* L.) leaves are rich in essential oils. The volatile oil contains phenolic compounds, monoterpenes and sesquiterpenes: thymol, carvacrol, *p*-cymene, *γ*-terpinene, and linalool [63], with antioxidant, anti-inflammatory, and antimicrobial activity. In traditional medicine, oregano has been used for gastrointestinal disorders—indigestions, stomachache, and diarrhea; respiratory diseases—asthma and bronchitis; menstrual disorders; and diabetes, due to its anti-bacterial and anti-inflammatory activity [64,65,66,67].

A remarkable property of oregano essential oils is their antiproliferative activity on adenocarcinoma gastric cell line [68]. In case–control studies, gastric cancer was associated with red and processed meat consumption [69]. *Origanum vulgare* essential oils demonstrated inhibitory effects on the growth of carbapenem-resistant Gram-negative bacteria [70], and antibacterial activity and synergistic effect with polymyxin B against multidrug-resistant *Acinetobacter baumannii* [71], whose development is favored by extensive antibiotic utilization, including animal treatments [72,73]. 

Oregano has a high content of antioxidants, according to a study conducted by Zheng and Wang, which makes it suitable for use as a natural antioxidant. Antioxidant compounds are phenolic acids and flavonoids, such as caffeic acid, rosmarinic acid, hispidulin, and apigenin, as well as carvacrol and thymol, components of the essential oil [74]. 

Fasseas et al. evaluated the antioxidant activity of meat treated with oregano and sage essential oils extracted by hydrodistillation. In this regard, the minced pork and beef were formulated in three samples as follows: homogenization with 3% (*w*/*w*) of either oregano essential oil or sage essential oil, and a control sample which did not contain essential oils. The samples thus obtained were stored at 4 °C in raw and cooked states (85 °C, 30 min), and the antioxidant activity was evaluated at 1, 4, 8, and 12 days of storage. Essential oils have led to a decrease in lipid oxidation, the role of antioxidants being affected by meat proteins and was significantly more important in cooked meat [75].

Plant extracts and essential oils including thyme, oregano (rich in thymol and carvacrol), rosemary, and sage are used to prevent the oxidation of meat products, in encapsulated form or in edible films, due to their high solubility, and for flavoring, due to their organoleptic properties [76].

The effect of oregano essential oil used in the formulation of an active coating used on fresh pork meat was studied. The essential oil was used as free oil, nanoemulsified or microencapsulated. All formulated samples showed a delay in the oxidation of lipids and oxymyoglobin, and the sensory profile was more appreciated as opposed to the control sample which did not contain oregano essential oil [44]. 

**Basil** (*Ocimum basilicum* L.) leaves contain many antioxidant and anti-inflammatory flavonoids such as quercetin, quercetin-3-*O*-diglycoside, querce-tin-3-*O*-β-d-galactoside, quercetin-3-*O*-*β*-d-glucoside, quercetin-3-*O*-*β*-d-glucoside-2″-gallate, quercetin-3-*O*-(2″-*O*-galloyl)-rutinoside, querce-tin-3-*O*-*α*-l-rhamnoside, isoquercetrin, kaempferol; carotenoids—*β*-carotene, *β*-cryptoxanthin, and lutein–zeaxanthin; polyphenols—rosmarinic acid, and chicoric acid (dicaffeoyltartaric acid); coumarin, aesculetin, and *p*-coumaric acid [77]. In traditional medicine, *Ocimum basilicum* has been used for the treatment of digestive disorders and demonstrates carminative, stimulant, antispasmodic, antidiarrheal, antibacterial, and anthelmintic effects [31]. It has also been used in treating vomiting, flatulence, dyspepsia, and gastritis [78]. *Ocimum basilicum* leaves contain caffeic acid, which demonstrates antioxidative and cancer chemopreventive properties [79]. *Ocimum basilicum* essential oils exhibited cytotoxic activity against human liver hepatocellular carcinoma cell lines (HEpG2) and nasopharyngeal cancer cell line (KB) [78]. In experimental models, *Ocimum basilicum* demonstrated anti-hyperglycemic potential, and antioxidant and nephroprotective effects in diabetic disease [80,81,82].

The antioxidant compounds in basil extracts with a role in antiradical activity are chlorogenic, *p*-hydroxybenzoic, caffeic, vanillic, and rosmarinic acids, as well as apigenin, quercetin, and rutin. Teofilović et al. performed various extractions with mixtures of ethanol–water (30%, 40%, 50%, 60%, 96% *v*/*v*), concentrated methanol (95% *v*/*v*), water (in presence and absence of light), dichloromethane, chloroform, and hexane over different periods of time (10 and 30 min), which exhibited antioxidant activity by the DPPH method with IC_50_ values between 0.22 ± 0.01 and 12.99 ± 0.87 g/mL for polar solvents and from 12.12 ± 0.54 to 20.49 ± 1.54 g/mL for non-polar solvents. Increasing the extraction time and polarity of the solvent improve the quality of the extracts in terms of phenolic compounds and antioxidant capacity [83].

Basil essential oil was added in various concentrations (0.062%, 0.125%, and 0.25%) to a beef burger to evaluate its natural antioxidant effectiveness. The results showed that the essential oil decreased the rate of lipid oxidation, and the effectiveness did not depend on the concentration [13].

**Marjoram** (*Origanum majorana* L.) leaf essential oils have flavonoids and terpenoids as the main active compounds [42,84]. These are excellently summarized in a review published by Bina [85]. Monoterpene hydrocarbons are represented by *α*- and *β*-pinene, *α*- and *β*-phellandrene, camphene, sabinene, limonene, *ρ*-cymene, *β*-ocimene, *γ*-terpinene, *α*-terpinene, terpinolene, carvone, and citronellol. Thymol, carvacrol, and linalool are other monoterpene compounds. Sweet marjoram essential oil contains phenolic compounds such as rosmarinic acid, sinapic acid, vanillic acid, ferulic acid, caffeic acid, and coumarinic acid, and phenolic glycosides such as arbutin, methyl arbutin, vitexin, and orientinthymonin [85]. Flavonoids such as hesperetin, kaempferol, and luteolin were also found in marjoram extracts and have vasoprotective effects, together with carvacrol and thymol [86]. Marjoram is traditionally used to treat respiratory and gastrointestinal diseases, high blood pressure, dysrhythmia, pains, and fatigue. Marjoram extracts demonstrated antioxidant, anti-inflammatory, anti-hyperglycemic, hypouricemic, anticancer, gastro-, nephro-, and hepatoprotective activity [85]. Acetylcholinesterase and tyrosinase inhibitory activities were also demonstrated, and support the antineurodegenerative effects of marjoram [87]. Essential oils and extracts from *O. majorana* exhibit antiparasitic, antifungal and antimicrobial activity against the Gram-positive species *Staphylococcus aureus*, *Enterococcus faecalis*, and *Streptococcus dysgalactiae*, and the Gram-negative species *Klebsiella pneumoniae* and *Pseudomonas aeruginosa* [88,89,90].

Marjoram essential oil has antioxidant properties through its ability to inhibit hydroxyl radicals. The major compounds of the oil are terpinen-4-ol (21.3%), *trans*-sabinene hydrate (15.5%), *γ*-terpinene (14.0%), and *α*-terpinene (8.9%). At a concentration of 0.05%, marjoram essential oil inhibited the formation of conjugated dienes by 50% and the generation of oxidized by-products of linoleic acid by 79.85% through its addition to an emulsion system with linoleic acid [91].

**Mint** (*Mentha piperita* L.) leaf essential oils mainly contain oxygenated monoterpenes, monoterpene hydrocarbons, sesquiterpene hydrocarbons, and oxygenated sesquiterpenes [92]. Quantitatively, the most abundant constituents are menthol and menthone. Menthofuran, menthyl acetate, iso-menthone 1,8-cineole, and the toxicity of pulegone should be remarked [93]. The traditional use of mint for treating fevers, colds, digestive diseases, infections, and throat inflammation has been supported by experimental studies. Pharmacological activities of mint demonstrated to date include antioxidant, anti-inflammatory, anticancer, antidiabetic, hepatoprotective, neuroprotective, and radio-protective activity [94]. In healthy adults, essential oil rich in menthol/menthone attenuated mental fatigue associated with extended cognitive task performance [95]. Antimicrobial, antiviral, antifungal, biopesticidal, and larvicidal activity has also been reported.

Mint extracts contain a significant number of phenolic compounds, and thus exert an important antioxidant activity. Their antioxidant activity can be compared with that of synthetic antioxidants [96].

In their study, Kanatt et al. treated lamb pulp with mint extract before it was irradiated (2.5 kGy). An amount of 0.05 g/100 g mint extract inhibited oxidation to some extent (0.6 mg MDA/kg as opposed to using a 0.1 g/100 g extract, which led to inhibition of 50% (0.4 mg MDA/kg) [97]. In another study, Biswas et al. treated ground pork meat with mint extract, and it obtained a good color stability compared with samples obtained with sodium nitrite [98].

**Tarragon** (*Artemisia dracunculus* L.) leaf essential oils are rich in phytochemicals including coumarins, isocoumarins, monoterpenoids, sesquiterpenoids, flavonoids, polyacetylenes, and alkaloids. Tarragon increases bile and gastric acid production, stimulates digestion, and has beneficial effects on gastritis [99]. Essential oil concentrated in coumarin derivatives showed remarkable anticoagulant activity, inducing a therapeutic value (2.34) for the international normalized ratio (INR) in vitro [100]. In muscle cell cultures derived from lean, overweight, and diabetic–obese subjects, bioactive compounds of *Artemisia dracunculus* L. improved insulin sensitivity [101]. Essential oils also exhibited strong antifungal activity [102].

The antioxidant capacity of tarragon (*Artemisia dracunculus*) essential oil (0.01–0.9%) was studied by Behbahani et al. and the results showed that a concentration of 0.9% has an antioxidant activity of 78.87%, similar to that of the synthetic antioxidant BHT (butylated hydroxytoluene) [102].

Nimse and Pal present carotenoids, antioxidant vitamins, hydroxycinnamic acids, flavonoids and terpenes as antioxidant compounds that can help prevent oxidation in meat and meat products [103]. The phenolic content of plant-derived materials has the most significant potential in terms of the antioxidant and antimicrobial activity [76]. 

**Coriander** (*Coriandrum sativum* L.) leaves have high contents of vitamin C, vitamin A, vitamin K, iron, manganese, thiamine, zinc, *β*-carotene, and anthocyanins, and are used for the treatment of iron and vitamin deficiencies or as potent antioxidants. Fruits and leaves of *Coriandrum sativum* are traditionally used for digestive diseases, nausea, vomiting, indigestion, and against worms [104]. In experimental models, coriander seeds prevented gastric mucosal lesions induced by ethanol due to the protective layer formed by its hydrophobic compounds and free radical scavenging activity of its antioxidant constituents such as flavonoids, coumarins, catechins, and terpenes [105]. *Coriandrum sativum* L. leaf essential oils contain natural antimicrobial compounds that can act against *Candida* spp. [106] and *Campylobacter jejuni* found in beef and chicken meat [21]. Previous studies have demonstrated the antioxidant and neuroprotective effect of *Coriandrum sativum* L. extracts on brain [107], the decrease in brain cholinesterase activity and serum total cholesterol levels, and memory improvement [108]. Extracts from the leaves and stems of *Coriandrum sativum* L. exhibited significant antihyperglycemic activity, and seed extracts normalized glycemia and decreased the elevated levels of insulin [109].

Šojić et al. [110] effectively investigated the addition of coriander essential oil (0.075–0.150 μL/g) to pork sausages which also contained different levels of sodium nitrite (0, 50 and 100 mg/kg). In addition to nitrite, coriander essential oil contributes to lower lipid oxidation due to its antioxidant potential due to terpenoid compounds. The essential oil contains linalool (835.2 mg/g), camphor (32.9 mg/g), *γ*-terpinene (32.8 mg/g), geraniol (16 mg/g), and (+)-limonene (6.2 mg/g) [111].

**Bay** (*Laurus nobilis* L.) leaf essential oils contain monoterpenes and monoterpenoids, sesquiterpenes and sesquiterpenoids, diterpenoids, phenyl propene derivatives, alcohols, carbonyls, and esters [112]. *Laurus nobilis* L. is used in the treatment of cancer, gastrointestinal disorders, epilepsy, rheumatic conditions, and several infectious diseases [113]. In an experimental model of ulcers induced by ethanol, bay leaf extracts demonstrated gastric mucosal protection correlated with antioxidant activity [114]. 

The antimicrobial activity of essential oils tested against *Escherichia coli*, *Staphylococcus aureus*, *Enterococcus faecalis*, *Pseudomonas aeruginosa*, and *Candida albicans* exhibited inhibitory effects and antioxidant activity [112]. 

Bay leaf essential oil has antioxidant and antibacterial properties in meat due to oxygenated monoterpenes and phenolic compounds, according to the study conducted by Ramos et al. [115]. Bay leaf essential oil has been used to treat packaged chicken in a microaerobic atmosphere where it has reduced oxidation and prolongs the smell of fresh meat, making it suitable for use as a natural preservative. Bornyl acetate, 1,8-cineole, *β*-myrcene, and carvacrol represents the principal components from bay leaf essential oil [116].

**Thyme** (*Thymbra capitata* (L.) Cav) leaf essential oils contain mainly monoterpenes, generally 10% carvacrol and 50% thymol, but also linalool, *α*-terpineol, camphor, caryophyllene, and *γ*-terpinene [117]. Flavonols (quercetin-7-*O*-glucoside), flavanones (naringenin) and flavones (apigenin), phenolic acids (*p*-coumaric, caffeic, rosmarinic, cinnamic, carnosic, ferulic, quinic, and caffeoylquinic acids), saponins, steroids, alkaloids, and tannins have also been described in thyme extracts [117,118,119]. In traditional medicine, thyme has been used as a sedative, a carminative, an additive for baths, or an infusion for the treatment of skin diseases [117,118]. Thyme extract might be an effective treatment of chronic respiratory diseases accompanied by the inflammation and hypersecretion of mucus [120]. Previous studies have reported that thymol demonstrates anti-inflammatory, anti-carcinogenic, and immunomodulatory properties, decreased serum lipids, visceral fat accumulation, and reduced blood pressure in experimental models [121,122]. Thyme essential oils have exhibited antifungal and antibacterial bioactivity against both food spoilage microflora and pathogenic microflora in vitro, including *B. subtilis*, *S. aureus*, *S. enteritidis*, and *P. aeruginosa* [123]. Thyme essential oils demonstrated antimicrobial activity against *Listeria monocytogenes* in vivo [124]. 

Due to the carvacrol present in the composition of thyme essential oil, it is distinguished by its antioxidant capacity. Thus, through their study on chicken breast in which thyme essential oil was added in a proportion of 0.5%, Fratianni et al. demonstrated that it reduced radical formation and lipid peroxidation and prolonged the shelf life of products [125]. In another study, Zengin and Baysal added thyme essential oil to minced beef that was stored for 9 days at 4 °C. Thyme essential oil showed a delaying effect in the oxidation process of lipids and color, and the sensory quality of the product was not affected by this addition [126].

## 4. Essential Oils and Aroma Compounds of Spicy and Aromatic Plants

Chemically, essential oils are a complex mixture of numerous bioactive chemical components, such as terpenes, terpenoids, and phenolics. Essential oils are synthesized by almost all plant organs, particularly the flowers, buds, leaves, seeds, stems, roots, and fruits [127]. Moreover, essential oils exhibit a very characteristic odor, and are therefore responsible for the specific scents that aromatic plants emit. These essential oils can be stored in epidermal cells, cavities, secretory cells of glandular trichomes. Numerous essential oils have the potential to be used as a food preservative for meat and meat products [128,129,130,131,132,133]. It should be highlighted that essential oils are generally accepted by consumers due to their high volatility and biodegradable nature.

**Rosemary**. Oxygenated monoterpenes (74.1%), represented mainly by 1,8-cineole (33.1%), camphor (18%), and borneol (7.95%), are the major terpenes of the Tunisian rosemary essential oil (Table 2). Monoterpene hydrocarbons constitute 21.6% of the oil, and α-pinene (10.16%) is the major compound of this class. The amount of oxygenated sesquiterpenes, represented only by caryophyllene oxide, was low (0.62%) [134].

According to the literature [44,135,136,137,138], the main primary components of the oil are: 1,8-cineole, α-pinene, camphor, verbenone, and borneol, whereas the secondary components are terpinen-4-ol, *α*-terpineol, *β*-caryophyllene, 3-octanol, geranyl acetate, and linalyl acetate. *Rosmarinus officinalis* L. volatile oil can be distinguished according to its 1,8-cineole, *α*-pinene, and camphor content. Additionally, certain essential oil compositions are dominated by myrcene [139].

The volatile fraction of *R. officinalis* differs for many of the chemotypes growing in the countries listed in Table 2, with regard to compounds, in the genus of components and their relative quantity. The observed differences may probably be due to different environmental and different chemotypes and the nutritional status of the plants, as well as other factors that can influence the oil composition. It is of interest to note the presence of *p*-cymene (44.42%) in very high percentages, which was distinctive of *R. officinalis* [140]. Monoterpenes constituted the major compounds of the oil (86.3%), whereas sesquiterpenes amounted to 11% [141].

**Mint**. The essential oil of *M. pullegium* is rich in pulegone (35.1%) and piperitenone (27.4%) (Table 2). However, none of these compounds were present in the essential oil of the other Mentha species, i.e., mint, whose main component is carvone (75.9%). Pulegone (48%) and menthone (41%) are described in the essential oil of the *M. pullegium* species [135]. According to Erich et al., the chemical composition of peppermint oil, as detailed in Table 2, is dominated by menthol (40.7%) and menthone (23.4%) [142]. The odor perceptions of peppermint essential oil include green, herbaceous, bitter, mint, or fresh. The amount of peppermint essential oil is influenced by geographical area, ripening time, or soil type [143].

**Thyme.** The major component identified from wild thyme was carvacrol (56.0%) [135]. Carvacrol (55.1%) and geraniol (43%) were the main components of the oils from *Thymus x citriodorus* “Archer’s Gold” and Thymus x citriodorus, respectively (Table 2). Among the other constituents, *p*-cymol, *β*-caryophyllene, geranial, limonene, and *γ*-terpinene were also characteristic of the oils, but smaller amounts were present. Thymol is the main component of oils from *Thymus vulgaris* and *Thymus serpyllum* (37.1% and 17%, respectively) [144].

**Coriandrum**. Essential oils from the fresh herb *Coriandrum sativum* L. accumulate during the growth of the plant. According to Wei et al., it is recommended to harvest the vegetative part prior to flowering because (*E*)-2-decenal, a potential irritant, is present at higher percentages in the preflowering and full flowering stages [145]. 

There are significant differences in the essential oil types from different parts of coriander. (*E*)-2-decenal is the dominant constituent in essential oils from coriander leaves: 31.28% and 61.86% of the essential oils from coriander stems [145]. Linalool was found to be the main constituent of essential oils from fully mature coriander fruits, seeds, and pericarps. Linalool and citronellol are the main components of coriander inflorescence essential oils from lower latitudes. The young vegetative organs in the seedling stages not only have a grassy odor, peculiar to coriander, but also a fresh green odor, highly suitable for seasoning [146].

**Sage** it is one of the most appreciated herbs due to its rich essential oils, which have antimicrobial, antifungal, and antimutagenic properties [128]. More than 120 components have been characterized in the essential oil prepared from aerial parts of *S. officinalis*. The main components of the oil include borneol, camphor, caryophyllene, cineole, elemene, humulene, ledene, pinene, and thujone [53].

**Basil.** Oxygenated monoterpenes and phenylpropanoids are the main compounds of the *Ocimum* genus. In different *O. basilicum* cultivars and chemotypes, linalool, eugenol, methyl chavicol, methyl cinnamate, methyl eugenol, and geraniol have been reported as the major components [147].

**Parsley.** In terms of essential oils, the major compound in parsley’s essential oils is myristicin, which has spicy, warm, and balsamic odors. The combined cleaning process with drying at different temperatures provided a greater reduction in the microorganisms in relation to separate processes. Drying did not change the oil yield in relation to the fresh plant [148].

**Tarragon.** According to Sobieszczanska et al., tarragon essential oil and its major compounds act against food-associated *Pseudomonas* spp. Tarragon essential oils are rich in other compounds, particularly methyl chavicol (tarragon and basil essential oils [149].

**Origanum** essential oils, together with sage, rosemary, marjoram, and thyme essential oils, are mainly composed of oxygenated monoterpenes (>40%) [150]. Generally, the *Origanum* species is characterized by the presence of two major biochemically related groups of compounds (aromatic monoterpenes such as *p*-cymene, thymol, carvacrol, their precursor *γ*-terpinene, and their derivatives; thujanes, such as sabinene, sabinene hydrate, and their derivatives) [151]. Thymol and carvacrol, which are present in high amounts in its essential oils (78–82%), are generally responsible for its antioxidant properties [128]. 

**Marjoram.** The results of marjoram essential oil analysis by Dimitra et al. gave a large number of constituents. Among them, 3-thujene (2.8%), *γ*-myrcene (3.8%), 2-carene (7.8%), 2-ethyl-mxylene (5.2%), 3-carene (10.4%), terpinen-4-ol (7.8%), sabinene hydrate (6.0%), R-terpineol (4.2%), and thymol (14%) were detected. Two chemotypes of *O. majorana* are reported in the literature: the cis-sabinene hydrate/tepinen-4-ol chemotype and the carvacrol/thymol chemotype [151].

**Bay.** The predominant flavor compound is 46% eucalyptol; essential oils of bay leaf induced human aryl hydrocarbon receptor activity by threefold, and its major constituent eucalyptol (46%) was inactive [152]. 

**Dill.** Terpenes were the most abundant volatiles detected in dill essential oils. Dill oil had a relatively limitary chemical profile, approximately equal volumes of carvone and d-limonene. Together, these accounted for 97.5% of the compounds identified by gas chromatography/mass spectroscopy [153].
plants-11-00960-t002_Table 2Table 2Variability in the chemical composition of EOs and aroma compounds of spices and aromatic plants used to prepare meat and meat alternatives.
OriginChemotypeOdor Perception ^a^ReferencesRosemary



*Rosmarinus officinalis* L.Mexico14.1% α-Pinene, 11.5% camphene, 12.0% β-pinene, 7.9% α-phellandrene, 8.6% 1,8-cineole, 3.4% 2-bornanone, 8.7% camphorPine, camphor, turpentine, resin, turpentine, turpentine, mint, spice, sweet[154]
Australia, USA, South Africa, Kenya, Nepal, and Yemen13.5%–37.7% α-pinene, 16.1%–29.3% 1,8-cineole, 0.8%–16.9% verbenone, 2.1%–6.9% (−)-borneol, 0.7%–7.0% camphor, 1.6%–4.4% racemic limonene.Pine, mint, sweet, camphor[137]
Brazil26.0% Camphor, 22.1% 1,8-cineol, 12.4% myrcene, 11.5% α-pineneCamphor, pine, turpentine, mint, sweet, balsamic, must, spice[136]
Portugal1.2% α-humulene, 7.2% α-terpineol, 35.4% verbenoneWood, oil, anise, mint[135]
Portugal16.6–29.5% Myrcene, 8.3–14.5% 1,8-cineol, 14.3–23.1% camphorMint, sweet, camphor, balsamic, must, spice[138]
Morocco37.4% α-pinene, 41–53% camphor, and 58–63% 1,8-cineolPine, turpentine, mint, sweet, camphor[155]
Tunisia20–46% 1,8-cineol, 8.5–30.2% camphor, 6.5–13% α-pinene, 4–25% borneolPine, turpentine, mint, sweet, camphor[134]
Algeria48.9% Camphor
[139]
Egypt52.8% 1,8-cineol, 11.9% camphor, 10.2% α-pinene, 7.5% borneolMint, sweet, camphor[156]
Lebanon19.1–25.1% 1,8-cineol, 18.8–38.5% α-pinenePine, turpentine, mint, sweet[157]
Turkey44.02% p-Cymene, 20.5% linalool, 16.62% γ-terpinene, 2.64% 1,8-cineolSolvent, gasoline, citrus, turpentine, mint, sweet, flower, lavender[140]
Greece24.1% α-Pinene, 14.9% camphor, 9.3% 1,8-cineol, 8.9% camphenePine, turpentine, camphor, mint, sweet, camphor[141]
Sardinia (Italy)23% α-Pinene, 16% borneol, 9.4% verbenone, 10.4% bornyl acetatePine, turpentine[158]
Spain18.2% Eucalyptol, 35.5% (−)-Camphor, 13.4% (−)-BornylacetateCamphor[150]
Spain, Morocco, and Tunisia5–21% camphor, 15-55% 1,8-cineole, 9–26% pinene, 1.5–5.0% borneol, 2.5–12.0% camphene, 1.5–5.0% limonene.Pine, turpentine, camphor, mint, sweet[40]Mint



*Mentha spicata*Portugal75.9% CarvoneMint[135]*Mentha pullegium*Portugal35.1% Pulegone, 27.4% PiperitoneAromatic, minty, green, herbaceous, bitter, mint, fresh*Mentha viridis*Morocco37.26% carvone, 11.82% 1,8-cineole, 08.72% Terpinen-4-ol(leaves)Citrus; herbaceous; fruity; sweet; vanilla; minty; pepper; spicy; woody, turpentine, nutmeg, must[159]*Mentha x piperita L.*USA40.7% menthol, 23.4% menthoneCool-minty, minty[142]Thyme



*Thymus serpyllum*Portugal56.0% carvacrol, 4.9% α-terpineol, 2.7% veridiflorolOil, anise, mint, sweet, green, herbal, fruity, tropical, minty[135]*Thymus vulgaris*Hungary37.1% thymol, 3.1% carvacrolHerb, spicy[144]*Thymus serpyllum*17% thymol, 2.3% carvacrolHerb, spicy*Thymus x citriodorus*0.8% thymol, 6.1% carvacrol, 43% geraniolHerb, spicy, rose, geranium*Thymus x citriodorus “Archer’s Gold”*0.7% thymol, 55.1% carvacrolHerb, spicy*Thymus mastichina*Spain75.4% m-thymol, 5.4% carvacrolHerb, spicy[150]*Thymus vulgaris*Greece4.3% γ-terpinene, 23.5% p-cymene, 2.2% carvacrol 63.6% thymolGasoline, turpentine, solvent, citrus[151]Coriander



*Coriandrum sativum L.*North India (Haldwani)62.1% linalool, 7.3% (2e)-dodecenal, 4.1% n-dodecanal, 4.1% α-pinene (inflorescence eo)Flower, lavender, green, fat, sweet, pine, turpentine[160]
Tunisia (Korba)86.1%, 91.1% and 24.6% linalool (fruit, seed, and pericarp)Flower, lavender[161]
Seoul, Korea23.11% cyclododecanol, 17.86% tetradecanal, 9.93% 2-dodecenal, 7.24% 1-decanol, 6.85% 13-tetradecenal, 6.54% 1-dodecanol, 5.16% dodecanal, 2.28% 1-undecanol, and 2.33% decanal (leaves)Mandarin, fat, soap, orange peel, tallow[162]
Austria60.5 % δ^3^-carene, 18.2% γ-terpinene,Orange peel, gasoline, turpentine, lemon, resin[150]Salvia



*Salvia officinalis*Mexico12.2% eucalyptol, 28.7% (−)-camphor, 12.6% (−)-bornylacetateCitrus, herbaceous; fruity, sweet, vanilla, minty, pepper, spicy, woody, camphor[150]
Italy20.16% α-thujone, 14.04% 1,8-cineole, 10.09% β-pinene (flower)Pine, resin, turpentine, mint, sweet[163]Basil



*Ocimum basilicum*The Island of Comoro77.9% methylchavicolLicorice, anise[150]*Ocimum basilicum cv. Keshkeni luvelu*Iranlinalool, 1,8-cineole, tau-muurolol, and α-cadinol (major compounds)Flower, lavender, mint, sweet[147]Parsley



*Petroselinum sativum*USA45.1% 4-methoxy-6-(2-propenyl)-1.3-benzodioxoleSpice, warm, balsamic[150]*Petroselinum crispum Mill.*Brazilapiole, myristicin (major compounds)Spice, warm, balsamic[148]Tarragon



*Artemisia dracunculus*Spain92.4% methylchavicolLicorice, anise, clove, spice, mint, turpentine[150]
Spain24.5% methyl eugenol, 19.3% β-phellandreneClove, spice, mint, turpentine
Origanum



*Thymbra capitata* (L.) Cav.
78.4% carvacrol, 10.9% m-thymolHerb, spicy[150]*Oroganum dictamus*Greece12.7% γ-terpinene, 9.9% p-cymene, 7.8% carvacrol, 63.3% thymolGasoline, turpentine, solvent, citrus[151]*Origanum vulgare L.*
46.84% thymol, 12.88% γ-terpineneGasoline, turpentine[163]Marjoram



*Organum majorana*Greece2.8% 3-thujene, 3.8% β-myrcene, 7.8% 2-carene, 5.2% 2-ethyl-mxylene, 10.4% 3-carene, 7.8% terpinen-4-ol, 6.0% sabinene hydrate, 4.2% β-terpineol, 14% thymol.Balsamic, must, spice, lemon, resin, wood, green, herb[151]Bay



*Laurus nobilis*USA46% eucalyptolCitrus, herbaceous, fruity, sweet, vanilla, minty, pepper, spicy, woody[152]Dill



*Anethum graveolens*
40% limonene, 44% caraway, 25% spearmint, 9% star anise.Lemon, orange[152]^a^ Odor descriptions according to the Flavornet (www.flavornet.org(accessed on 3 January 2022) and Pherobase databases (www.pherobase.com(accessed on 3 January 2022)).

## 5. Conclusions

The increasing interest of consumers for most natural foods, natural flavors, natural antioxidants, natural antimicrobial substances is often linked to their bioactive compounds; spices and aromatic plants are some of the richest sources of phytochemicals with demonstrated biological activities. The most commonly used spice and aromatic plants in the meat and meat analogues industry are parsley, dill, basil, oregano, sage, coriander, rosemary, marjoram, tarragon, bay, thyme, and mint. As shown in this paper, they improve meat preparation in terms of flavor, as well as antimicrobial and antioxidant activity. Due to their bioactive compounds, they also have beneficial effects on the consumer health. Although studies of the effects of these herbs and spices on meat analogues are limited, their use is highly recommended due to the flavor improvements, as well as consumers health benefits.

## Figures and Tables

**Table 1 plants-11-00960-t001:** The effect of different parts of spicy and aromatic plants in meat products.

Plant Name	Plant of Origin	Part of the Plant	Form Used	Quantity Used	Product Added	Effect	Reference
Parsley	*Petroselinum crispum* Hoffm.	Parsley stems	Extract powder	4.29 g/kg	Mortadella-type sausage	Inhibition of *L. Monocytogenes* and microbial spoilage during storage time	[10]
Dill	*Anethum* *graveolens* L.	Seeds	A water-soluble polysaccharide named AGP1 was isolated from seeds of *Anethum graveolens*	0.3% (*g*/*g*)	Turkey sausages	Preservative AGP1 replaced ascorbic acid, reduced lipid peroxidation, preserved pH and color and improved bacterial stability during cold storage at 4 °C for 12 days	[11]
Basil	*Ocimum* *basilicum* L.	Leaves	Essential oils	9.0 μl/ml	Fermented sausages	Reduction in mold growth; antifungal protection	[12]
				0.062–0.25%	Beef burger	Antioxidant and antibacterial activity	[13]
Oregan	*Origanum* *vulgare* L.	Dried leaves from Chile leaves	Essential oils	0.230–0.690 mg/ml	Sausage	Bacteriostatic effect; antimicrobial activity	[14]
				6.25–100 μl ml^−1^	Poultry meat products	Bio-preservative; antimicrobial activity against *Staphylococcus aureus*	[15]
Sage	*Salvia officinalis* L.	Leaves	Ground sage leaves	0.05–0.15%	Chinese style sausage	Improve the oxidative stability of Chinese-style sausage antioxidant activity	[16]
			Essential oils	0.2 și 0.5%	Chicken fat	Effective in respect of hydrolytic rancidity	[17]
		Sage tea processing by product	Essential oil	0.05 μl/g–0.1 μl/g	Fresh pork sausages	Significant antioxidative and antimicrobial activities	[18]
			Herbal dust	0.05 μl/g–0.1 μl/g	Fresh pork sausages	Significant antioxidative and antimicrobial activities	[18]
Coriander	*Coriandrum* *sativum* L.	Leaves	Essential oils	0.01%	Italian salami	Reduction in lipid oxidation by increasing the shelf life of the product	[19]
				0.02%	Stored ground beef	Inhibiting the development of unwanted sensory changes and the growth of Enterobacteriaceae	[20]
				0.075–0.150 μl/g	Cooked pork sausages	Improved oxidative stability	[21]
Rosemary	*Rosmarinus* *officinalis* L.	Leaves	Water extract	0.4% rosemary spice and 0.6% nitrite pickling salts	Beef sausages	Microbial inhibition;Rosemary spice can substitute nitrite pickling salt	[22]
			Extract in powder	0.025–0.05%	Fermented goat meat sausage	Oxidative stability (antioxidant activity)	[23]
			Essential oils	0.2 și 0.5%	Chicken fat	Antioxidant activity; effective in respect of hydrolytic rancidity	[17]
Marjoram	*Origanum* *majorana* L.	Leaves	Essential oils	6.25-100 μl ml^−1^	Poultry meat products	Bio-preservative; antimicrobial activity against *Staphylococcus aureus*	[15]
Tarragon	*Artemisia* *dracunculus* L.	Leaves	Essential oils	0.062–0.25%	Beef burger	Natural preservative, flavor enhancer in meat; antioxidant activity and antibacterial effects; anti-*staphylococcus aureus* activity	[13]
				0.1% (*v*/*w*)	Frankfurter type sausages	Improving sensory properties	[24]
Bay	*Laurus nobilis* L.	Leaves	Essential oils	10% (*v*/*v*)	Fresh lamb meat	Natural preservative; antibacterial activity	[25]
				0.05 g–0.1 g/100 g	Fresh Tuscan sausage	Antibacterial activity; improve the safety and shelf life	[26]
Thyme	*Thymbra* *capitata* (L.) Cav	Leaves	Essential oils	0.01–3% [*v*/*w*]	Minced beef meat	Antibacterial (from *Escherichia coli, Salmonella typhimurium, Staphylococcus aureus, Pseudomonas aeruginosa*)	[27]
Mint	*Mentha piperita* L.	Leaves	Essential oils	20, 40 and 60 ppm	Cooked sausage	Nitrite partial replacement with *Mentha piperita* essential oil proved oxidative, microbial, and sensory properties	[28]

## Data Availability

Not applicable.

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
