# Peer review of "Spicy and Aromatic Plants for Meat and Meat Analogues Applications"

_plants, 2022, doi:10.3390/plants11070960_

Round 1
Reviewer 1 Report
See enclosed file

Author Response
The Editor
Plants
Subject: Submission of revised manuscript No. Plants: 1656965
Dear Sir
It is stated that I want to submit revised article entitled, “Spicy and Aromatic Plants for Meat and Meat Analogues Ap-plications” for publication in your esteemed Journal. We are highly thankful to referees whose comments helped in improving this manuscript. We have revised the entire manuscript for proper flow of the information. These are the reviewers’ observations which are addressed in the point by point rebuttal file and also incorporated the same in the text.
Below is response to editor and referee comments:
Reviewer 1
Comment: The manuscript is quite interesting and well written.
Seversl mistakes must be corrected.
Table 1 Salvia Rosmarinus L. It is not correct. Salvia rosmarinus Schleid. but the accepted name is Rosmarinus officinalis L.
Table 1 Thymus capitatus Hoff. et Link The correct accepted name is Thymbra capitata (L.) Cav.
line 110 β italics
line 112 β italics
line 146 Salvia rosmarinus Schleid. but the accepted name is Rosmarinus officinalis L.
line 151 '-O-(O-acetyl)-β-D-glucuronide
line 154 Salvia rosmarinus Schleid. but the accepted name is Rosmarinus officinalis L.
line 205 α-thujone, β-thujone
line 212 Salvia officinalis italics
line 228 D-limonene
line 241 CH2Cl2
line 245 p-cymene, γ-terpinene
line 265 4°C
lines 279-281 quercetin-3-O-diglycoside, quercetin-3-O-β-D-galactoside, quercetin-3- O-β-D-glucoside, quercetin-3- O-β-D-glucoside-2”-gallate, quercetin-3-O-(2”-O-galloyl)-rutinoside, querce-tin-3-O-α-L-rhamnoside,
line 282 β-carotene
line 283 β-cryptoxanthin
line 284 p-coumaric acid
line 295 p-hydroxybenzoic
line 300 IC50
line 310-312 α- and β-pinene, α- and β-phellandrene, camphene, sabinene, limonene, ρ-cymene, β-ocimene, γ-terpinene, α-terpinene,
lines 328-329 trans-sabinene hydrate (15.5%), γ-terpinene (14.0%), and α-terpinene
line 336 the toxicity of pulegone should be remarked
line 370 β-carotene
line 388 γ-terpinene
line 404 β-myrcene
line 406 Thymus capitatus Hoff. et Link The correct accepted name is Thymbra capitata (L.) Cav.
line 408 γ-terpinene quercetin-7-O-glucoside
line 409 p-coumaric
line 446 α-pinene
line 447 α-terpineol, β-caryophyllene
line 448 Rosmarinus officinalis
line 449 α-pinene
line 456 ρ-cymene
line 470 Thymus x citriodorus italics
line 471 p-cymol, β-caryophyllene, geranial, limonene, and γ-ter
line 473 Thymus vulgaris and Thymus serpyllum
line 477 (E)-2-decenal
line 480 (E)-2-decenal
line 507 ρ-cymene
line 508 γ-terpinene
line 512 γ-myrcene
Table 2 Insert the botanical authority for all the species. Some are missing. Check italics in all the chemical names as previously stated. Origanum Õulgare = Origanum vulgare . Thymus capitatus Hoff. et Link = Thymbra capitata (L.) Cav.
Authors response: Thank you for encouraging comments. All mistakes have been corrected.
Reviewer 2
Comment: The manuscript is well written and all the information laid out for the interest of the readers. However, some important information is missing such as the sensory discussion on the sensory evaluation and the effect on the texture related to the addition of spicy and aromatic plants.
What are the challenges involved in the acceptance of the products to the general public?
How cost effective is it to commercial this on large scale products?
Authors need to cite the following review " The Flavor of Plant-Based Meat Analogues " and follow some of the schematics added to the current review to make it interesting for the readers.
Authors response: Thank you for encouraging comments.
As mentioned in the manuscript: "The aim of this paper is to present the latest information on the bioactive antioxidant and antimicrobial properties of the most used herbs and spices (parsley, dill, basil, oregano, sage, coriander, rosemary, marjoram, tarragon, bay, thyme, mint) used in the meat and meat analogues industry or proposed to be used for meat analogues" and “This review aims to gather recent information on spicy and aromatic plants used to prepare meat and meat alternatives”.
Deasemnea scopul acestui Special Issues este " Spicy and Aromatic Plants".
Also the purpose of this Special Issues is "Spicy and Aromatic Plants".
If we were to introduce another chapter with information on sensory evaluation, texture issues, costs and cost-effectiveness, we would overlap with the manuscripts mentioned below, which were published a few months ago and are not part of this review.
https://www.mdpi.com/2304-8158/10/3/600
https://www.mdpi.com/2304-8158/9/9/1334
https://www.mdpi.com/2304-8158/10/4/801
https://www.mdpi.com/2304-8158/10/2/260
https://www.sciencedirect.com/science/article/pii/S030881462101445X
Regarding the suggestion to quote the review "The Flavor of Plant-Based Meat Analogues", I'm sorry but I didn't find any manuscript with this title.
Best regards,
Marc (Vlaic) Romina Alina et al.

Reviewer 2 Report
The manuscript is well written and all the information laid out for the interest of the readers. However, some important information is missing such as the sensory discussion on the sensory evaluation and the effect on the texture related to the addition of spicy and aromatic plants.
What are the challenges involved in the acceptance of the products to the general public?
How cost effective is it to commercial this on large scale products?
Authors need to cite the following review " The Flavor of Plant-Based Meat Analogues " and follow some of the schematics added to the current review to make it interesting for the readers.
Author Response

(The authors gave the same response as above.)
